# [Re] Latent Embedding Feedback and Discriminative Features for Zero-Shot Classification

1 ## Reproducibility Summary

*In this study, we show our results and experience during replicating the paper titled "Latent Embedding Feedback and Discriminative Features for Zero-Shot Classification". We have updated the model for the recent PyTorch version. We were able to reproduce both the quantitative and qualitative results, as reported in the paper which includes inductive, finetuning and reconstruction of the original images from synthesized features. The authors have open-sourced their code for inductive setting. We have implemented the codes for the finetuning setting and reconstruction of the images.*

**Scope of Reproducibility**

`TF-VAEGAN` [11] proposes to enforce a semantic embedding decoder (SED) at training, feature synthesis and classification stages of (generalized) zero-shot learning. They introduce a feedback loop, from SED for iteratively refining the synthesized features during both the training and feature synthesis stages. The synthesized features, along with their corresponding latent embeddings from the SED are then transformed into the discriminative features and utilized during the classification stage to reduce ambiguities among the categories.

**Methodology**

As the `TF-VAEGAN` method was available in PyTorch 0.3.1, we ported the entire pipeline to PyTorch 1.6.0 along with implementing the finetuning and reconstruction codes from scratch. Our implementation is based on the original code and on the discussions with the authors.[1] Total training times for each method ranged from 2-8 hours on Caltech-UCSD-Birds [16] (CUB), Oxford Flowers [12] (FLO), SUN Attribute [13] (SUN), and Animals with Attributes2 [17] (AWA2) on a single NVIDIA Tesla V100 GPU. Further details are presented in Table 3.

**Results**

We were able to reproduce the results quantitatively on all the four datasets as reported in the original paper as well as reconstruct the original images from the generated features.

**What was easy**

The authors' code was well written and documented, and we were able to reproduce the preliminary results using the documentation provided with the code. The authors were also extremely responsive and helpful via email.

**What was difficult**

The feature reconstruction codes from [5, 10, 4] are not available in PyTorch. Therefore, we had to implement it in PyTorch along with a hyperparameter search to get the images. We also performed a hyperparameter search for getting the finetuning results.

**Communication with original authors**

We reached out to the authors a few times via email to ask for clarifications and additional implementation details.

---

[1]Our implementation can be found at https://anonymous.4open.science/r/ZSL_Generative-D397/.

Submitted to ML Reproducibility Challenge 2020. Do not distribute.

# 1 Introduction

Zero-shot learning (ZSL) is a challenging vision task that involves classifying images into new "unseen" categories at test time, without having been provided any corresponding visual example during training. In the generalized variant, the test samples can further belong to seen or unseen categories. Most recent work in ZSL and GZSL recognition [18, 6, 19, 8, 9] are based on Generative Adversarial Networks (GANs), where a generative model is learned using the seen class feature instances and the corresponding class-specific semantic embeddings. Feature instances of the unseen categories, whose real features are unavailable during training, are then synthesized using the trained GAN and used along with the real feature instances from the seen categories to train zero-shot classifiers in a fully-supervised setting. In this reproducibility report, we study the proposed work by Narayan et al. [11] in detail, which consists of implementing the architecture described in the paper, running experiments, reporting the important details about certain issues encountered during reproducing, and comparing the obtained results with the ones reported in the original paper. We report our numbers on seen accuracy, unseen accuracy and Harmonic mean in Table 4.

# 2 Scope of Reproducibility

The core finding of the paper is that utilizing semantic encoder decoder (SED) at *all* stages (i.e training, feature synthesis, and classification) of a VAE-GAN based ZSL framework helped to obtain absolute gains of 4.6%, 7.1%, 1.7%, and 3.1% on CUB [16], FLO [12], SUN [13], and AWA2 [17], respectively for generalized zero-shot (GZSL) object recognition on comparing to baseline. To achieve this, the authors introduced the following two methods:

- A feedback module that transforms the latent embeddings of the SED and modulates the latent representations of the generator for utilizing SED during training and feature synthesis.

- A discriminative feature transformation, used at the classification stage, utilizes the latent embeddings of SED along with respective features.

In order to provide effective re-implementation, we make sure that the quantitative results are reproduced with marginal errors, which might have caused due to porting the codes to a newer PyTorch, TorchVision and CUDA Toolkit version. We also assure that our visual results look similar to those presented in the original paper.

# 3 Methodology

Firstly, we ran the authors' publicly available code on all the four CUB, FLO, SUN and AWA2 datasets to get the preliminary results. Secondly, We ported the publicly available code to a recent PyTorch version and made sure that to get on-par results to the original code. Thirdly, We used the authors shared fine-tuned features to train the inductive method code to get fine-tuned results along with the hyperparameter search. Lastly, we implemented the code for reconstructing the images from the synthesized features. Furthermore, we integrated WandB [2] library to the training loop to track our experiments during training.

## 3.1 TF-VAEGAN

`TF-VAEGAN` architecture is an VAE-GAN based network with an additional semantic decoder (SED) $Dec$ at both the feature synthesis and (G)ZSL classification stages. The authors introduced a feedback module $F$ which is used during training and features synthesis stage along with the Decoder $Dec$. We employ the same architecture as that of the authors in which the VAE-GAN consists of an Encoder $E$, Generator $G$ and Discriminator $D$. Real features of seen classes $x$ and the semantic embeddings $a$ are input to $E$ which gives the parameters of a noise distribution as the output. The KL divergence ($\mathcal{L}_{KL}$) loss is applied between these parameters and a zero-mean unit-variance Gaussian prior distribution. The network $G$ synthesizes the features $\hat{x}$ using noise $z$ and embeddings $a$ as inputs. Further, a binary cross-entropy loss $\mathcal{L}_{BCE}$ is integrated between the synthesized features $\hat{x}$ and the original features $x$. The discriminator $D$ takes either $x$ or $\hat{x}$ along with embeddings $a$ as input and outputs a real number, thus determining whether the input is real or fake. The WGAN loss $\mathcal{L}_W$ is used at the output of $D$ that learns to distinguish between the real and fake features. The architecture design focuses on the integration of an additional semantic embedding decoder (SED) $Dec$ at both the feature synthesis and (G)ZSL classification stages. The paper proposes to use a feedback module $F$, along with $Dec$, during the training and feature synthesis. Both $Dec$ and $F$ collectively address the objectives of enhanced feature synthesis and reduces vagueness among categories during classification. The $Dec$ takes either $x$ or $\hat{x}$ and reconstructs the embeddings $\hat{a}$. It is trained using a cycle-consistency loss $\mathcal{L}_R$. The learned $Dec$ is subsequently used in the (G)ZSL classifiers. The feedback module $F$ transforms the latent embedding of $Dec$ and feeds it back to the latent representation

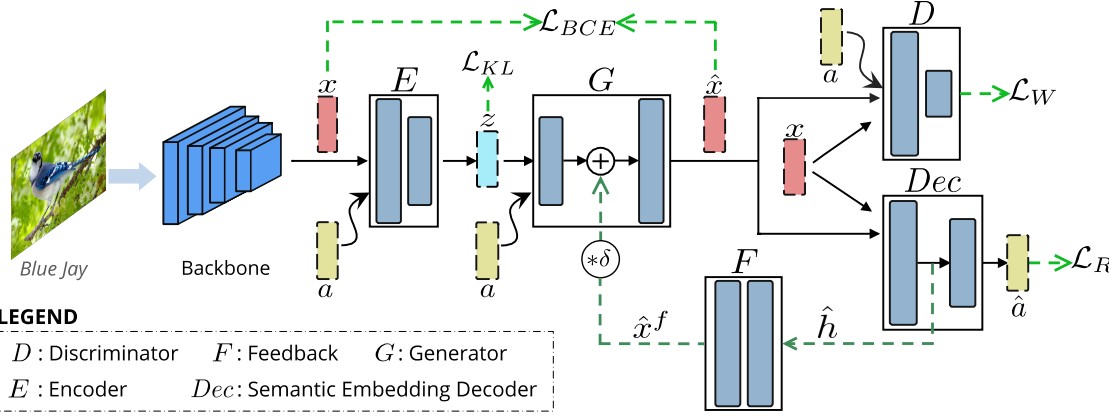

Figure 1: Representation of the `TF-VAEGAN` architecture. Obtained from the paper [11]. A seen class is fed to a backbone network to extract seen visual features $x$ which is then input to the encoder $E$ with the corresponding embeddings $a$. Latent code $z$ is been generated by the encoder, which is then combined with embeddings $a$ and fed to the generator $G$ for synthesizing features $\hat{x}$. The discriminator $D$ is learnt to distinguish between real and synthesized features $x$ and $\hat{x}$, respectively. A binary cross-entropy loss ($\mathcal{L}_{BCE}$) and the KL divergence ($\mathcal{L}_{KL}$) is been used to train the VAE (comprising $E$ and $G$). A WGAN loss ($\mathcal{L}_W$) is used to train both $G$ and $D$ forming the GAN. Authors introduced a semantic embedding decoder $Dec$ (Sec. 3.1.1) to reconstruct the embeddings $\hat{a}$ using a cycle-consistency loss ($\mathcal{L}_R$). To transform the latent embedding $\hat{h}$ of $Dec$ and feed it back to $G$, for iteratively refining $\hat{x}$ authors introduced a feedback module $F$ (Sec. 3.1.3).

of generator $G$ to achieve improved feature synthesis. The SED $Dec$ and feedback module $F$ are described in detail in Sec. 3.1.1 and 3.1.3.

### 3.1.1 Semantic Embedding Decoder

The authors introduce a semantic embedding decoder $Dec : \mathcal{X} \rightarrow \mathcal{A}$, which reconstructs the semantic embeddings $a$ from the generated features $\hat{x}$. This helps to enforce a cycle-consistency on the reconstructed semantic embeddings thus ensuring that the generated features are transformed to the same embeddings that generated them. As a result, semantically consistent features are obtained during feature synthesis. The cycle-consistency of the semantic embeddings was achieved using the reconstruction loss, $\ell_1$ as:

$$\mathcal{L}_R = \mathbb{E}[||Dec(x) - a||_1] + \mathbb{E}[||Dec(\hat{x}) - a||_1]. \tag{1}$$

The loss formulation for training the proposed `TF-VAEGAN` can be defined as:

$$\mathcal{L}_{total} = \mathcal{L}_{vaegan} + \beta \mathcal{L}_R, \tag{2}$$

where $\beta$ is a hyper-parameter for weighting the decoder reconstruction error. The authors utilized SED at *all* three stages of VAE-GAN based ZSL pipeline: training, feature synthesis and classification.

### 3.1.2 Discriminative feature transformation

Next, the authors introduce a discriminative feature transformation scheme to effectively utilize the auxiliary information in semantic embedding decoder (SED) at the ZSL classification stage. The generator $G$ learns a *per-class* "single semantic embedding to many instances" mapping using only the seen class features and embeddings. The SED was trained using only the seen classes but learns a *per-class* "many instances to one embedding" inverse mapping. Thus, the generator $G$ and SED $Dec$ are likely to encode complementary information of the categories. Here, the authors propose to use the latent embedding from SED as a useful source of information at the classification stage. The detailed overview of the architecture is presented in Figure 2a.

### 3.1.3 Feedback Module

Lastly, the authors introduce a feedback loop for iteratively refining the feature generation during both the training and the feature synthesis phase. The feedback loop was added between the semantic embedding decoder $Dec$ and the generator $G$, via the incorporated feedback module $F$ (see Fig. 1. The proposed module $F$ enables the effective

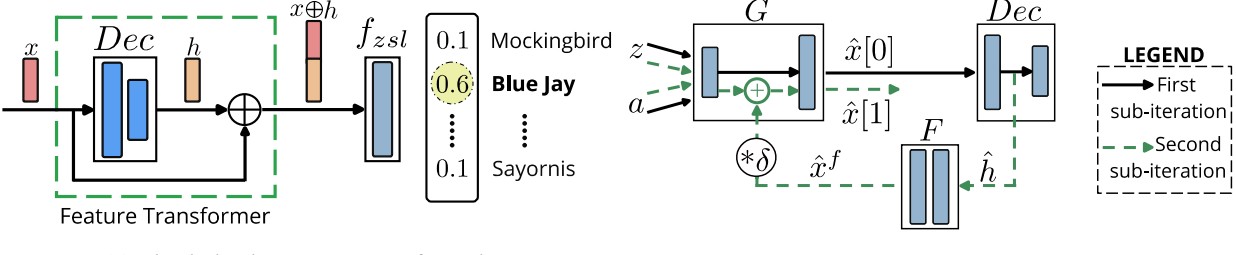

(a) Discriminative Feature Transformation          (b) Feedback Module

Figure 2: (a) **Integration of SED:** Taken from the original paper [11]. Authors used the $Dec$ at the ZSL/GZSL classification stage. The input seen visual features $x$ are concatenated ($\oplus$) with the respective latent embedding $h$ from SED for feature transformation. These transformed discriminative features are then used for ZSL/GZSL classification. (b) **Feedback module brief:** Taken from the original paper [11]. Firstly, the initial features $\hat{x}[0]$ are synthesized using the generator $G$. These initial features are then passed through the $Dec$. Secondly, the latent embedding $h$ from $Dec$ are transformed to $\hat{x}^f$ using the module $F$, which represents the feedback to $G$. Then, the enhanced features $\hat{x}[1]$ are synthesized by the generator $G$ using the same $z$ and $a$ along with the feedback $\hat{x}^f$.

utilization of $Dec$ during both training and feature synthesis stages. Let $g^l$ denote the $l^{th}$ layer output of $G$ and $\hat{x}^f$ denote the feedback component that additively modulates $g^l$. The feedback modulation of output $g^l$ is given by,

$$g^l \leftarrow g^l + \delta\hat{x}^f, \tag{3}$$

where $\hat{x}^f = F(h)$, with $h$ as the latent embedding of $Dec$ and $\delta$ controls the feedback modulation. The authors based their feedback loop on [15], which introduces a similar feedback module but for the task of image super-resolution. The authors make necessary modifications in the feedback module in order to use it for zero-shot recognition as a naive plug-and-play of the module provides sub-optimal performance for zero-shot recognition. The detailed overview of the feedback module is given in Figure 2b.

## 3.2 Datasets

We evaluated the `TF-VAEGAN` method on four standard zero-shot object recognition datasets: Caltech-UCSD-Birds [16] (CUB), Oxford Flowers [12] (FLO), SUN Attribute [13] (SUN), and Animals with Attributes2 [17] (AWA2) containing 200, 102, 717 and 50 total categories, respectively. CUB contains 11,788 images from 200 different types of birds annotated with 312 attributes. SUN contains 14,340 images from 717 scenes annotated with 102 attributes. FLO dataset has 8189 images from 102 different types of flowers without attribute annotations. Finally, AWA2 is a coarse-grained dataset with 30,475 images, 50 classes and 85 attributes. We use the same splits as used in the original paper for AWA2, CUB, FLO and SUN insuring that none of the training classes are present in ImageNet [3]. Statistics of the datasets are presented in Table 3.

## 3.3 Finetuning

For fine-tuning results, we used the same approach as discussed in the original paper. We used the original ResNet-101 [7] that is pre-trained on ImageNet-1k[3] and fine-tune the last layer of ResNet-101 on the seen training dataset of CUB, AWA2, FLO, and SUN respectively. We further use the fine-tuned layer to extract seen visual features which is used for training the `TF-VAEGAN` method.

## 3.4 Reconstruction

We follow a strategy similar to [11] and used a upconvolutional neural network to invert feature embeddings to the image pixel space. A generator consisting of a fully connected layer followed by 5 upconvolutional blocks was used for reconstruction task. An upconvolutional block was build using an Upsampling layer, a 3x3 convolution, BatchNorm and ReLU non-linearity. We reconstructed the image with an image size of 64x64. Then, a discriminator is used to processes the image through 4 downsampling blocks, the feature embedding is sent to a linear layer and spatially replicated and concatenated with the image embedding, and this final embedding is passed through a convolutional and sigmoid layer to get the probability that the sample is real or fake. We used a L1 loss between the ground truth image and the inverted image, along with a perceptual loss, by passing both images through a pre-trained ResNet101, and then we calculated an L2 loss between the feature vectors at conv5_4 and average pooling layers. To improve the image quality of our image,

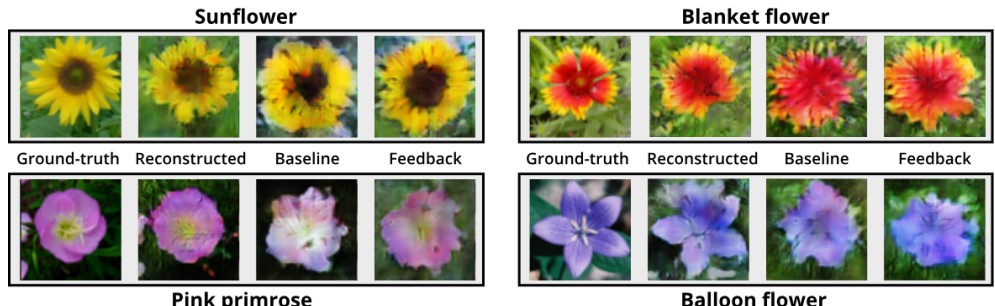

Figure 3: Feature reconstruction results. Taken from the original paper [11]

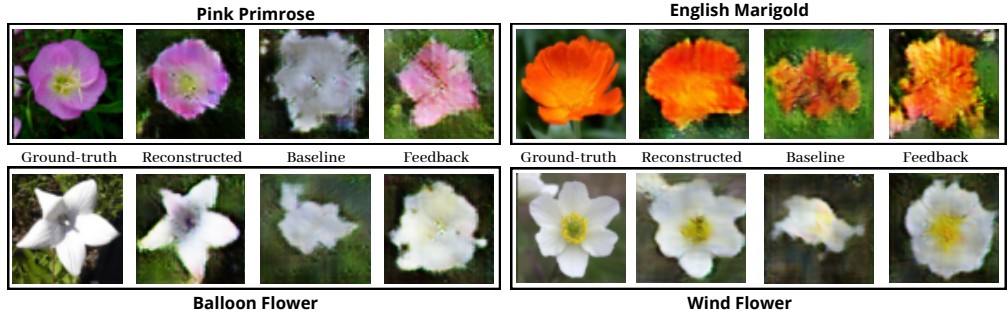

Figure 4: Ours reproduced feature reconstruction results. From the above figure, we were able to show that the `TF-VAEGAN` method is able to generate more visually similar features for the Ground-truth. The feedback provided from the `TF-VAEGAN` method also provides features which are more similar to the Ground-truth in terms of colours and shape.

we used an adversarial loss by feeding the image and feature embedding to a discriminator. We train this model on all the real feature-image pairs of the 102 classes of FLO [12] dataset, and use the trained generator to invert images from synthetic features. Reconstructed images can be seen in Figure 4.

# 4 Implementation details

## 4.1 Training strategy

We follow the same training strategy as that of the paper for training discriminative feature transformation. First, the feature generator $G$ and the semantic embedding decoder $Dec$ are trained. Then, $Dec$ is used to transform the features (real and synthesized) to the embedding space $\mathcal{A}$. The latent embeddings from $Dec$ are then combined with the respective visual features. Let $h_s$ and $\hat{h}_u \in \mathcal{H}$ denote the hidden layer (latent) embedding from the $Dec$ for inputs $x_s$ and $\hat{x}_u$, respectively. The transformed features are represented by: $x_s \oplus h_s$ and $\hat{x}_u \oplus \hat{h}_u$, where $\oplus$ denotes concatenation. In proposed `TF-VAEGAN` method, the transformed features are used to learn final ZSL and GZSL classifiers as,

$$f_{zsl} : \mathcal{X} \oplus \mathcal{H} \to \mathcal{Y}^u \qquad \text{and} \qquad f_{gzsl} : \mathcal{X} \oplus \mathcal{H} \to \mathcal{Y}^s \cup \mathcal{Y}^u. \tag{4}$$

As a result, the final classifiers learn to distinguish categories using transformed features properly. The authors used semantic embedding decoder $Dec$ as the input, as it was used to reconstruct the class-specific semantic embeddings from features instances. In original paper $G$ and $F$ are trained alternately [15] to utilize the feedback for improved feature synthesis. In the proposed alternating training strategy, the generator training iteration is unchanged. However, during the training iterations of $F$, two sub-iterations are performed:

- *First sub-iteration*: The noise $z$ and semantic embeddings $a$ are input to the generator $G$ to yield an initial synthesized feature $\hat{x}[0] = G(z, a)$, which is then passed through to the semantic embedding decoder $Dec$.

- *Second sub-iteration*: The latent embedding $\hat{h}$ from $Dec$ is input to $F$, resulting in an output $\hat{x}^f[t] = F(\hat{h})$, which is added to the latent representation (denoted as $g^l$ in Eq. 3) of $G$. The same $z$ and $a$ (used in the first sub-iteration) are used as input to $G$ for the second sub-iteration, with the additional input $\hat{x}^f[t]$ added to the

latent representation $g^l$ of generator $G$. The generator then outputs a synthesized feature $\hat{x}[t+1]$, as,

$$\hat{x}[t+1] = G(z, a, \hat{x}^f[t]). \tag{5}$$

The refined feature $\hat{x}[t+1]$ is input to $D$ and $Dec$, and corresponding losses are computed (Eq. 2) for training. In practice, the second sub-iteration is performed only once. The feedback module $F$ allows generator $G$ to view the latent embedding of $Dec$, corresponding to current generated features. This enables $G$ to appropriately refine its output (feature generation) iteratively, leading to an enhanced feature representation. The detail overview of the model architecture is given in Table 1.

## 4.2 Experimental Setup

In this study, we have followed the same training procedures for all the settings, as described in the original paper. The parameters for all training settings can be found in the configuration file in our GitHub repository. Our implementation is open-sourced, and can be accessed at https://anonymous.4open.science/r/ZSL_Generative-D397/

**Visual features and embeddings:** Following the same approach as discussed in the paper, we extracted the average-pooled feature instances of size 2048 from the ImageNet-1k [3] pre-trained ResNet-101 [7]. For semantic embeddings, we use the class-level attributes for CUB (312-d), SUN (102-d) and AWA2 (85-d). For FLO, fine-grained visual descriptions of image are used to extract 1024-d embeddings from a character-based CNN-RNN [14].

## 4.3 Hyperparameters details

The discriminator $D$, encoder $E$ and generator $G$ are implemented as two-layer fully-connected (FC) networks with 4096 hidden units. The dimensions of $z$ and $a$ are set to be equal ($\mathbb{R}^{d_z} = \mathbb{R}^{d_a}$). The semantic embedding decoder $Dec$ and feedback module $F$ are also two-layer FC networks with 4096 hidden units. The input and output dimensions of $F$ are set to 4096 to match the hidden units of $Dec$ and $G$. We used to same activation function as used by the authors, therefore we used LeakyReLU activation with a negative slope of 0.2 everywhere, except at the output of $G$, where a *Sigmoid* activation is used for applying the BCE loss. The network is trained using the Adam optimizer with $10^{-4}$ learning rate. Final ZSL/GZSL classifiers are single fully-connected layer networks with output units equal to number of test classes. Hyper-parameters $\alpha$, $\beta$ and $\delta$ are set to 10, 0.01 and 1, respectively. The gradient penalty coefficient $\lambda$ is initialized to 10 and WGAN is trained, similar to [1]. We also did a hyperparameter search for the SUN dataset for the Finetune-inductive setting. Detailed results can be seen in the Table 2.

| Class Name | Module Name | Input Features | Output Features | Non-linearity |
|---|---|---|---|---|
| Encoder | FC 1 | 2360 | 4096 | Leaky ReLU(0.2) |
| | FC 2 | 4096 | 624 | Leaky ReLU(0.2) |
| | Linear Means | 624 | 312 | - |
| | Linear Log Vars | 624 | 312 | - |
| Generator | FC 1 | 624 | 4096 | Leaky ReLU(0.2) |
| | FC 2 | 4096 | 2048 | Sigmoid |
| Discriminator D1 | FC 1 | 2360 | 4096 | Leaky ReLU(0.2) |
| | FC 2 | 4096 | 1 | - |
| Feedback | FC 1 | 4096 | 4096 | Leaky ReLU(0.2) |
| | FC 2 | 4096 | 4096 | Leaky ReLU(0.2) |
| Attribute Decoder | FC 1 | 2048 | 4096 | Leaky ReLU(0.2) |
| | FC 2 | 4096 | 312 | Sigmoid |

Table 1: Details of the model implementation for the CUB [16] dataset. For other datasets, values of input features and output features of FC 1 and FC 2 respectively of the Encoder class changes accordingly. Similarly, both input features and output features of Linear Means and Linear Log Vars changes for the encoder class. For the Generator and Discriminator D1, only input features changes on varying the dataset. For the Attribute Decoder, only output features of the FC 2 changes.

| GAN_lr | Decoder_lr | Feedback_lr | a1 | a2 | zsl accuracy | unseen accuracy | seen accuracy | H |
|---|---|---|---|---|---|---|---|---|
| 0.00001 | 0.00001 | 0.00001 | 0.1 | 0.01 | 63.3 | 37.5 | 47.7 | 38.5 |
| 0.0001 | 0.0001 | 0.00001 | 0.1 | 0.01 | 64.3 | 42.2 | 47.0 | 44.4 |
| 0.0001 | 0.00001 | 0.00001 | 0.1 | 0.01 | 65.0 | 44.3 | 46.7 | 45.5 |
| 0.0001 | 0.00001 | 0.0001 | 0.1 | 0.01 | 64.3 | 42.6 | 49.3 | 45.7 |
| 0.0001 | 0.00001 | 0.00001 | 0.01 | 0.01 | 64.6 | 41.8 | 50.7 | 45.9 |
| 0.0001 | 0.00001 | 0.00001 | 0.1 | 0.01 | 64.6 | 42.4 | 50.0 | 45.9 |
| 0.0001 | 0.00001 | 0.000001 | 0.1 | 0.01 | 64.5 | 42.0 | 50.8 | 46.0 |
| 0.0001 | 0.00001 | 0.00001 | 0.1 | 0.1 | 65.0 | 42.3 | 50.8 | 46.1 |
| 0.0001 | 0.00001 | 0.000001 | 0.01 | 0.1 | 66.2 | 41.5 | 51.3 | 46.0 |

Table 2: Detailed results of the hyperparameter search for the SUN [13] dataset for the fine-tuned inductive setting. We observe on changing the values of the lr of GAN, Decoder, and Feedback we were able to replicate the *H* with minimum marginal difference of 0.2.

| Dataset | Attributes | $y^s$ | $y^u$ | Training Time (h) | Memory (GB) |
|---|---|---|---|---|---|
| CUB [16] | 312 | 100 + 50 | 50 | 4 | 2.6 |
| FLO [12] | - | 62 + 20 | 20 | 5.5 | 3.1 |
| SUN [13] | 102 | 580 + 65 | 72 | 7 | 2.6 |
| AWA2 [17] | 85 | 27 + 13 | 10 | 2.75 | 2.6 |

Table 3: CUB, SUN, FLO, AWA2 datasets, in terms of number of attributes per class (Attributes), number of classes in training + validation ($y^s$) and test classes ($y^u$). Total training time and memory consumption in terms of GB taken by each dataset is also reported in the above table.

## 4.4 Computational Requirements

All the experiments were run on the NVIDIA TESLA V100 with 32 GPU memory. A breakdown total training time taken by each datasets is provided in Table 3.

# 5 Results

We have implemented the model from scratch by following the descriptions provided in the original paper. We were able to replicate the claimed results of `TF-VAEGAN` by referring to the published code. Overall, our implementation of the `TF-VAEGAN` achieved relatively close ZSL accuracy and Harmomic mean in Inductive and Finetuned-Inductive settings for all the four datasets CUB [16], FLO [12], SUN [13] and AWA2 [17] with a marginal difference. Also, we were able to generate similar looking reconstructed images from the features, thus showing the effectiveness of the `TF-VAEGAN` feature synthesis stage. In Table 4, we report the original and our reproduced results with the paper's and our results on the two training setting Inductive and Finetune-Inductive.

**Caltech-UCSD-Birds::** Our implementation was able to replicate the results reported in the original paper with a marginal difference of $0.5 - 0.7\%$.

**Animals with Attributes2::** We were able to out-perform Generalized Zero-shot learning metric for seen classes with a difference of 1% and was able to replicate the others with marginal difference ranging from $0.2 - 1.0\%$. We saw a decrement in the performance for Zero-shot learning with a difference of 0.7%.

**Oxford Flowers::** Our implementation was able to out-perform Generalized Zero-shot learning metrics for unseen classes and harmonic mean with a difference of 0.6% and 0.1% respectively. We were also able to replicate the other results reported in the original paper with a marginal difference of $0.5 - 1.0\%$.

**SUN dataset::** We were able to improve Generalized Zero-shot learning metric for seen classes with a difference of 0.8%. We were also able to replicate others results from the original paper with a marginal difference of $0.4 - 1.9\%$.

| Dataset | Model | Zero-shot Learning | | | | Generalized Zero-shot Learning | | | | | | | | | | | |
|---|---|---|---|---|---|---|---|---|---|---|---|---|---|---|---|---|---|
| | | CUB T1 | FLO T1 | SUN T1 | AWA2 T1 | CUB | | | FLO | | | SUN | | | AWA2 | | |
| | | | | | | u | s | H | u | s | H | u | s | H | u | s | H |
| Inductive | Paper | 64.9 | 70.8 | 66.0 | 72.2 | 52.8 | 64.7 | 58.1 | 62.5 | 84.1 | 71.7 | 45.6 | 40.7 | 43.0 | 59.8 | 75.1 | 66.6 |
| | Ours | 64.4 | 70.3 | 65.5 | 71.5 | 52.2 | 64.0 | 57.5 | 63.1 | 83.1 | 71.8 | 43.7 | 41.5 | 42.6 | 58.8 | 76.1 | 66.4 |
| Finetune-Inductive | Paper | 74.3 | 74.7 | 66.7 | 73.4 | 63.8 | 79.3 | 70.7 | 69.5 | 92.5 | 79.4 | 41.8 | 51.9 | 46.3 | 55.5 | 83.6 | 66.7 |
| | Ours | 73.8 | 75.8 | 66.1 | 73.0 | 64.6 | 77.8 | 70.5 | 70.6 | 91.5 | 79.7 | 41.5 | 51.3 | 46.0 | 57.5 | 84.1 | 68.3 |

Table 4: Performance of our implementation on the different datasets on CUB, FLO, AWA2, and SUN. We compare our results to the paper's results and we were able to replicate the reported numbers in the original paper.

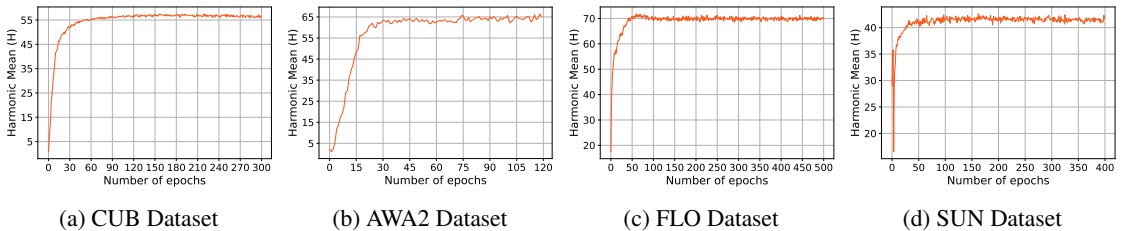

(a) CUB Dataset  (b) AWA2 Dataset  (c) FLO Dataset  (d) SUN Dataset

Figure 5: Harmonic mean curve for all the four datasets CUB, AWA2, FLO, and SUN.

**Feature Visualization:** To qualitatively assess the feature synthesis stage, we used the same approach as mentioned in the original paper and trained an upconvolutional network to invert the feature instances back to the image space by following a similar strategy as in [4, 19]. The model is trained on all real feature-image pairs of the 102 classes of FLO dataset [12]. The comparison between the original paper and ours reconstruction on Baseline and Feedback synthesized features on four example flowers are shown in Fig. 3 and Fig. 4 respectively. For each flower class, a ground-truth (GT) image along with three images inverted from its GT feature, Baseline and Feedback synthesized features, respectively are shown. Generally, inverting the features synthesized by the Feedback Module yields an image that is semantically closer to the GT image than the Baseline synthesized feature, suggesting that the Feedback module improves the feature synthesis stage over the Baseline, where no feedback is present.

## 6 Discussion

We found that the proposed feedback module in the VAE-GAN model helps modulate the generator's latent representation, improving the feature synthesis. By enforcing the generation of semantically-consistent features at all stages, the authors were able to outperform previous zero-shot approaches on four challenging datasets. The qualitative results generated with our replication are similar to those shown in the paper. Thus strengthening the authors' claim of highly-effective feedback module. As per out replicated quantitative results, we affirm that our implementation of `TF-VAEGAN` is consistent to the one provided by the authors. Overall, the paper and the provided code were sufficient for replicating the results on Inductive and Finetune-Inductive settings. For re-implementing the model from scratch, we have ported the code to a relatively new PyTorch version, and ended up with a **comparative** performance with the ones in the paper on all settings. Lastly, to provide an insight for run-time, we use the same hardware used by the authors, a Tesla V100 GPU card, on which we showing the runtimes on the four datasets.

**Recommendations for reproducibility:** Overall, the paper was clearly written and it was easy to follow the explanation and reasoning of the experiments. We ran into several obstacles while making a fairly-old environment of PyTorch 0.3.1. For reproducing the quantitative results, we had to assure that most, if not all, training/evaluation details were true to the experiments in the paper. We are extremely grateful to the original authors who gave swift responses to our questions. Nevertheless, it would have been easier to reproduce the results with latest PyTorch version compatibility. We hope our report and published code help future use of the paper.

**Recommendations for reproducing papers:** We recommend communicating early with the original authors to determine undisclosed parameters and pin down the experimental setup. Lastly, for reproducing training processes in particular, we suggest checking how training is progressing in as many different ways as possible. In our process, this involved looking at the progression of $H$ i.e Harmonic mean between GZSL and ZSL, examining training curves for individual loss function terms, both of which helped us pinpoint our issues.

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
