# OpenReview forum: "[Re] Latent Embedding Feedback and Discriminative Featuresfor Zero-Shot Classification"
_ML_Reproducibility_Challenge/2021/Fall — Reject_

### Official Review · Reviewer_McF4 · 2022-03-01
**Reproducibility of "Latent Embedding Feedback and Discriminative Features for Zero-Shot Classification"**

**Rating:** 6
**Confidence:** 3

**Review:**

The submission is based on reproducing the results from the paper "Latent Embedding Feedback and Discriminative Features for Zero-Shot Classification". The authors were able to reproduce both the quantitative and qualitative results from the original paper. The work also provides a more updated PyTorch version implementation of the code and extends the publicly available codebase to two other settings - finetuning setting and reconstruction of images. Overall, the document is well written and goes into details of the architectures and different parameters used in the original paper being reproduced. The authors of this work also reached out to the authors of the work being replicated and were able to gain insights on the important parameters and training procedure choices. Some of the other positives of this work include:

1. PyTorch implementation of some of the parts of the work being replicated which were not originally available in PyTorch. This should help other researchers pick up the code which is all in PyTorch and be able to build on the top of it.
2. Clear explanation of the methodology and approach used in the work being replicated. The authors go into details of different modules of the work and explain the different choices made in them.
3. The authors have tried to use the exact same training procedure and hyperparameters in the original paper to make sure that a fair reproducibility effort is being made.
4. WandB logging has also being integrated which can always help in better monitoring of logs.

Some of the areas in which the authors might even further improve their work:
1. More exhaustive hyperparameter tuning for all 4 datasets. Although, some of it has been done for a subset of datasets, a more thorough sweep might help in covering even those ranges which were not tried in the original work. This might also reveal some more insights into the choices made in the original work.
2. Adding more ablation studies would also be helpful. There is ample discussion on the original work's architecture choices and different components, and more experiments focusing on specific ablations could shed more light on the impact of the different choices.

Overall, the document is well written and the authors' contribution in validating the reproducibility of results from the paper should be a valuable contribution to the community.

---

### Official Review · Reviewer_Xhgw · 2022-03-28
**Succeeding in reproducing the core results but lack of motivations or insights**

**Rating:** 5
**Confidence:** 3

**Review:**

The paper comes with the well-written concise reproducibility summary, and in the introduction part the authors stated that they have re-implement the architecture described in the paper from scratch which does not match what they stated at the beginning of the reproducibility summary where they have adapted the model to the recent pytorch version and re-implemented the fine-tuning setting’s part and reconstruction of the images’ part. I have checked the code repo attached to this paper where I did not see lots of modifications to the original paper’s authors’ code. Therefore, I think the major contribution of the paper does not come with re-implementing the architecture of original paper from scratch but providing extra missing details to the original author’s original repo (e.g. code for reconstructing the images, fine-tuning, providing configurations files to run experiments easily, upgrading model to be able to run with recent pytorch version). The authors did a good job with respect to the communication with the original authors’ part and hyperparameter search part, and they are able to reproduce core qualitative and quantitative results in the paper to validate the main idea of the paper. They provided additional and intermediate training results, and compared the results to that of the original paper.  However, the paper lacks providing extra insights to the original paper (does not provide results beyond the paper). Also no ablation study seems to be provided.

---

### Meta-Review · Program_Chairs · 2022-04-09

**Recommendation:** Reject
**Confidence:** 5

**Metareview:**

The reviewers note that the submission would benefit from additional hyperparameter tuning, ablation, and motivation.  The submission is not accepted.

---

### Decision · Program_Chairs · 2022-04-09

Reject